# Study of Dynamic Evolution of the Shear Band in Triaxial Soil Samples Using Photogrammetry Technology

Yi Xia [1], Chunmei Mu [1,2,*], Wenjie Li [1], Kai Ye [1] and Haojie Wu [1]

1   School of Civil and Architectural Engineering, Guilin University of Technology, Guilin 541004, China
2   Guangxi Key Laboratory in Geotechnical Mechanics and Engineering, Guilin 541004, China
*   Correspondence: mchm@glut.edu.cn

**Abstract:** In order to avoid the influence of end restraint on triaxial stress and strain measurements, this study combines photogrammetry and computer technology to apply to the unconsolidated undrained test of triaxial soil samples. The novel method can establish an intuitive shear band evolution model of soil samples and help attain a clear and intuitive shear band evolution law. On the basis of the conventional triaxial test, this new method needs to paste the RAD (Ringed Automatically Detected) code points on the rubber film, plexiglass cover and brackets and then take pictures of the RAD code points in the loading process in a surrounding manner, with which the 3D shape of the restored soil samples can be determined. After eliminating the influence of refraction, the coded point cloud coordinates measured in the experiment can be used to calculate the axial deformation and radial deformation of the soil sample in the deformation process and determine the local stress–strain curve and three-dimensional displacement field diagram of the soil sample. The test shows that the new method can clearly determine the peak value and inflection point of the stress–strain curve in the middle region of soil samples, enabling it to reflect the shear change process of the soil sample more accurately. In addition, the displacement field can be used to directly observe the formation, development and penetration process of the shear band in soil samples.

**Keywords:** digital image; triaxial soil sample; shear band; shear band evolution

## 1. Introduction

The formation and evolution of the soil shear zone always draw much research attention in geotechnical engineering and can lead to slope landslide and subgrade sliding failure [1–3]. The traditional triaxial test that is usually used to test the soil mass performance is known to have limitations. Specifically, the assumption that the soil sample is isotropic and uniform deformation has to be made in the test [4–8]. Therefore, the stress–strain curve obtained from the test reflects the change of the soil sample as a whole unit and cannot monitor the local deformation of the soil sample at each stage, making it difficult to judge the shear band information of the soil sample. In practice, the influence of end restraint on soil usually results in more significant deformation in the middle of the soil mass, indicating that the conventional triaxial test still needs to be improved [9–11].

In the past, a great deal of research effort has been dedicated to developing different methods such as the double-wall cell [12,13] and volume controllers [14,15] for soil non-uniform deformation measurement during triaxial testing. Chen [16], Belmokhtar [17] and others successfully deduced the local deformations of the triaxial soil sample using the axial and radial deformations measured by the displacement sensors arranged on its surfaces. However, determining the formation process of the shear band by only relying on a small strain sensor could be very rough because the disturbance of measuring instruments to soil samples is difficult to avoid. CT technology was used by Alikarami [18] to observe the shear failure process of sand in a triaxial compression test. Cheng [19] and others further improved the CT technology and completed the scanning reconstruction of each

section of the soil sample in the triaxial test and captured the trend of strain localization. However, the expensive equipment cost and cumbersome operation significantly limit the wide application of this method in conventional soil tests. With the development of computer technology, a photogrammetric method based on digital images has become a good alternative. Sachan [20] successfully reconstructed the shape of a triaxial soil sample and calculated its local strain at different loading stages by taking photos to analyze the silhouette of the soil sample in the triaxial test. However, eliminating the image distortion caused by the pressure chamber during measurements seems to be problematic when applying this method. In order to avoid this problem, Salazar [21,22] proposed a strategy of taking pictures of triaxial soil samples with several cameras at different angles. Although such a method can obtain the deformation information of soil samples, it brings the problem of a cumbersome image acquisition system, so there is still a large space for improvement. Rui [23,24] successfully documented shear zone formation in the form of images in a shear test. Shen [25] and Wang [26] separately analysed the microstructure of coral sand and calcareous sand through digital image processing. The team led by Shao Longtan at Dalian University of Technology [27–29] applied the full surface digital image measurement system to the triaxial test to determine the strain distribution on the surface of the soil sample and provided some accurate data on the study of the shear band of the soil sample. However, this method requires a significant transformation of the pressure chamber, which is expensive and inconvenient to promote.

Based on digital image technology, this work proposes a measurement method that combines traditional photogrammetry with computer image processing technology. This method mainly includes two steps as follows. First, RAD (Ringed Automatically Detected) code points need to be pasted on the surface of the instrument and soil samples. Then, a camera is needed to take a 360° surround shot of the soil sample and transfer the shot film to the computer, where the PM (PhotoModeler Scanner) software will be adopted to post-process the obtained images and determine the deformation data of different areas of the soil sample.

## 2. Theory of 3D Model Reconstruction

### 2.1. Establishment and Transformation of the Coordinate System

In the present work, four coordinate systems are used to establish the mapping relationship between 2D image points and 3D coordinate points, including the world coordinate system, pixel coordinate system, pixel physical coordinate system and camera coordinate system. Among them, the world coordinate system needs to be established first to unify the three-dimensional information of each point on the soil sample surface. To establish this coordinate system, we pasted several columns of RAD on the left and right loading bars and triaxial apparatus surface in the middle, with the specific location as shown in Figure 1, and then used the lowest coding point of the left loading bar as the origin to establish an *O-XYZ* coordinate system as shown in the figure. The distance between the left and right loading bars was measured by a vernier caliper and input into the PM software to define the scale of the world coordinate system.

The images in the computer are stored in the form of pixel stacks. During data processing, PM software first sorts out the pixel array relationship of the images and sets the pixel coordinate system (*mAn*) with point A at the upper left corner of the image as the origin, as shown in Figure 2. Then, the software counts the number of pixels between left and right marker points and uses the prescribed scale to calculate the real size of a single pixel. Finally, the software establishes the image's physical coordinate system (*x′Ay′*) at point A. The transformation between the pixel coordinates and the image physical coordinates can be realized using formula (1), where $(m_1, n_1)$ is the coordinate of the space point in the pixel coordinate system, $(x_I', y_I')$ is the coordinate of the space point I in the pixel physical coordinate system, $(M, N)$ is the pixel unit of the image plane, and $(F_x, F_y)$ is the actual physical length and width of a single image.

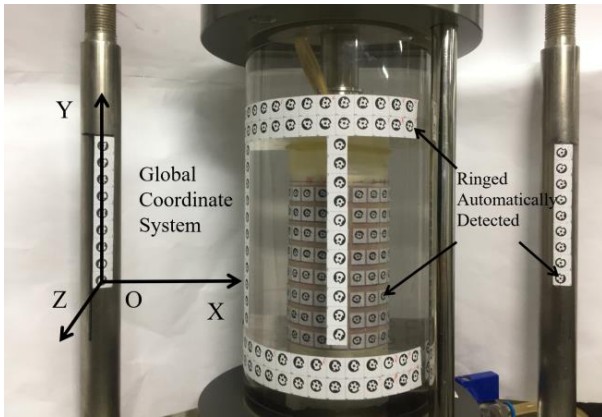

**Figure 1.** Diagram of the modified instrument.

$$\begin{bmatrix} x_I' \\ y_I' \end{bmatrix} = \begin{bmatrix} F_x/M & 0 \\ 0 & F_y/N \end{bmatrix} \begin{bmatrix} m_1 \\ n_1 \end{bmatrix} \tag{1}$$

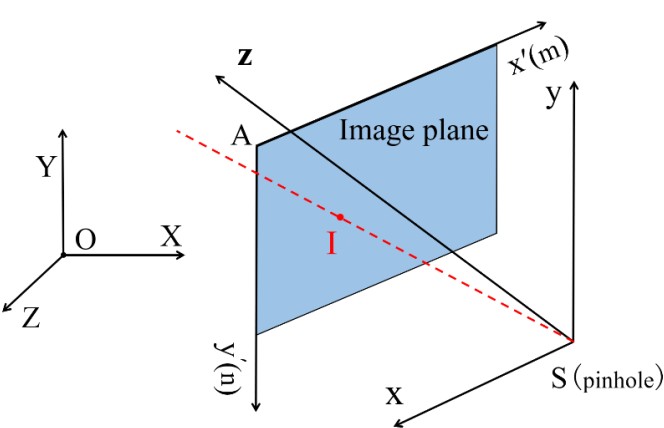

**Figure 2.** A sketch showing the coordinate transformation.

In order to transform the image pixel information from the physical coordinate system to the camera's local three-dimensional coordinate system ($S$-$xyz$), PM software uses the camera imaging principle to restore the angular relationship between the marked point and the camera and establishes the camera's local three-dimensional coordinate system with the camera's pinhole point S as the origin. The coordinate system transformation method is shown in Formula (2).

$$\begin{bmatrix} x_I \\ y_I \\ z_I \end{bmatrix} = \begin{bmatrix} x'_I - p_x \\ y'_I - p_y \\ -f \end{bmatrix} \tag{2}$$

In Equation (2), ($x_I$, $y_I$, $z_I$) is the coordinate of space point I in the camera coordinate system, $p_x$ and $p_y$ are the image physical coordinates from the camera optical center to the projection point of the image unit along the optical axis, and $f$ is the distance from the camera optical center to the image plane.

The association between the camera coordinate system and the world coordinate system adopts Equations (3) and (4), which were proposed by Li et al. [30–32]. In these equations, ($X_I$, $Y_I$, $Z_I$) refers to the coordinates of space point I in the world coordinate system, $R(\omega, \varphi, \kappa)$ is the rotation matrix, $\omega$, $\varphi$, $\kappa$ are the rotation angles of $X$, $Y$ and $Z$ axis of the camera coordinate system, respectively. The translation matrix ($X_s$, $Y_s$, $Z_s$) is used

to shift the coordinate origin of the camera coordinate system to the origin of the world coordinate system.

$$\begin{pmatrix} X_I \\ Y_I \\ Z_I \end{pmatrix} = R(\omega, \varphi, \kappa) \begin{pmatrix} x_1 \\ y_1 \\ z_1 \end{pmatrix} + \begin{pmatrix} X_S \\ Y_S \\ Z_S \end{pmatrix} \tag{3}$$

$$R(\omega, \varphi, \kappa) = \begin{pmatrix} \cos\kappa\cos\varphi & -\sin\kappa\cos\varphi & \sin\varphi \\ \cos\kappa\sin\omega\sin\varphi + \sin\kappa\cos w & -\sin\kappa\sin\omega\sin\varphi + \cos\kappa\cos\omega & -\sin\omega\cos\varphi \\ -\cos\kappa\cos\omega\sin\varphi + \sin\kappa\sin\omega & \sin\kappa\sin\varphi\cos\omega + \cos\kappa\sin\omega & \cos\omega\cos\varphi \end{pmatrix} \tag{4}$$

### 2.2. Determination and Correction of Depth Information

The propagation path of light in the triaxial test is: water → organic pressure chamber → air, during which the refraction effect will distort the obtained image of the soil sample [8,33]. A method to correct this is using the three-dimensional refraction correction model proposed by Xia et al. [34,35], which adopts the ray tracing method to correct the error of the image. The correction formula is shown in (5), which is used to determine the exact position of the imaging object point in three-dimensional space, where $n_a$ and $n_c$ are the refractive index of air and pressure chamber materials, $\vec{i_1}$ is the direction vector of the incident light, and $\vec{r_1}$ is the direction vector of light after refraction.

$$\vec{r_1} = \frac{n_a}{n_c}\vec{i_1} - \left( \frac{n_a}{n_c}\left( \vec{i_1} \cdot \vec{n_1} \right) + \sqrt{1 - \left( \frac{n_a}{n_c} \right)^2 \left( 1 - \left( \vec{i_1} \cdot \vec{n_1} \right)^2 \right)} \right) \vec{n_1} \tag{5}$$

The transformed coordinates can only estimate the azimuth information of the image shooting. In order to obtain the depth information of two-dimensional image points, we introduced the least square method as shown in Figure 3 [36,37], whose mathematic expression is given in Equation (6), with which we combined images from different angles to determine the minimum distance points from each optical path. In Equation (6), $(X_{Ci}, Y_{Ci}, Z_{Ci})$ is the three-dimensional coordinate of the $i$th point of the same marking point that intersects point $C_i$ on the inner surface of the glass cover, $(\alpha_{r2i}, \beta_{r2i}, \gamma_{r2i})$ represents the cosine of the refracted light $C_iP$ in the pressure chamber, $(X_p, Y_p, Z_p)$ is the P coordinate of the coding point obtained from the one-time shooting information, and the calculated S coordinate is considered as the "real" coordinate of coding point P.

$$\sum_{i=1}^{n} d_i^2 = \sum_{i=1}^{n} \begin{bmatrix} X_P - X_{C_i} \\ Y_P - Y_{C_i} \\ Z_P - Z_{C_i} \end{bmatrix}^T \begin{bmatrix} X_P - X_{C_i} \\ Y_P - Y_{C_i} \\ Z_P - Z_{C_i} \end{bmatrix} - \left\{ \begin{bmatrix} X_P - X_{C_i} \\ Y_P - Y_{C_i} \\ Z_P - Z_{C_i} \end{bmatrix}^T \begin{bmatrix} \alpha_{r2i} \\ \beta_{r2i} \\ \gamma_{r2i} \end{bmatrix} \right\}^2 \quad (n \geq 3) \tag{6}$$

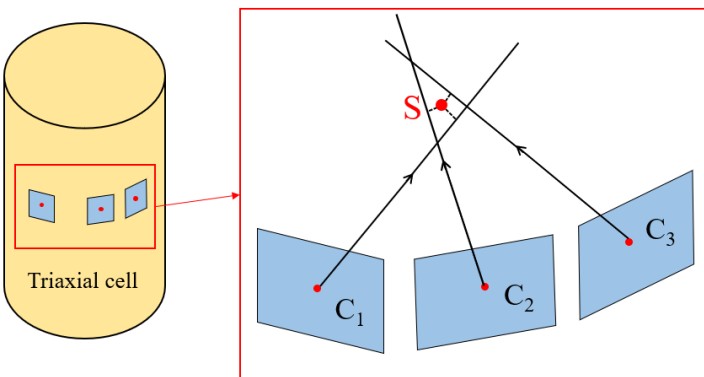

**Figure 3.** Multi-camera calibration model.

## 3. Experiment Scheme

### 3.1. Soil Sample

Natural red clay in Guilin urban area was selected as the research object, whose basic physical and mechanical property properties are summarized in Table 1. The soil is reddish brown with a high plastic limit and liquid limit. In order to better observe the formation of the shear band in the soil, we used the compacted sample with the maximum dry density [38]. With reference to Standard for Soil Test Methods (GB/T50123-2019), the static pressure sampling method was used to prepare soil samples with a dry density of 1.4 g/cm$^3$, 1.5 g/cm$^3$ and water content of 23% for standard triaxial tests, with a height of 80mm high and a diameter of 39.1 mm.

**Table 1.** Properties of the soil used in the present work.

| Property | Value |
| --- | --- |
| Natural moisture content (%) | 28 |
| Maximum dry density (g/cm$^3$) | 1.60 |
| Optimum moisture content (%) | 23 |
| Specific gravity | 2.65 |
| Liquid limit | 56.5 |
| Plastic limit | 31.8 |
| Plastic index | 24.7 |

### 3.2. Test Procedure

Different from the traditional triaxial test, this test adopted the photogrammetry as the main measurement method and used the strain control method to conduct the unconsolidated undrained (UU) test with the confining pressures of 100 kPa, 150 kPa and 200 kPa. The accuracy of the whole photogrammetric system had been verified before tests. Specific steps of the experiment referred to the work of Wu [39]. Each soil sample was wrapped with a rubber film, on which eight lines of coding points were pre-marked, with an interval of 10 mm between lines, as shown in Figure 4. The test shear rate was set as 0.2 mm/min. Before loading, we took a group of pictures around the soil sample as the initial data. During the test, each time the displacement increased by 1 mm, we paused the loading and took another group of pictures until the total displacement reached 14 mm. After the test, the photos were transferred to the PM software for calculation, with which the true three-dimensional coordinates of all camera optical centers, coding points on the surface of the pressure chamber and coding points on the surface of the soil sample were obtained after eliminating the refraction error.

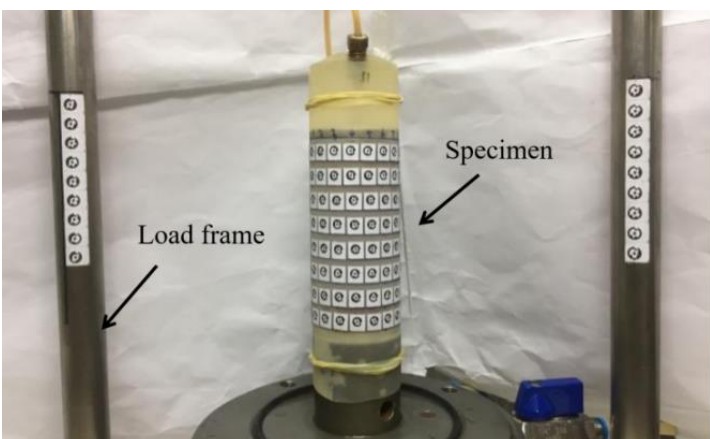

**Figure 4.** Lay-out of soil sample base coding points.

## 4. Results and Discussion

### 4.1. Processing and Analysis of Soil Sample Deformation Data

The area correction formula used in conventional triaxial tests for the strain calculation assumes that the soil sample is subject to uniform deformation in all directions. However, in actual tests, due to the existence of end constraints, the soil sample experiences uneven deformation, so the deformation data obtained cannot accurately reflect the shape of the soil sample. The image measurement technology used in this test can monitor the strain of each part of the soil sample without affecting the conventional triaxial test. Considering the influence of end restraint on specimen deformation, the test soil is divided into three regions (A, B and C) from top to bottom, as shown in Figure 5. The plane coordinate data at the same axial height in each region are fitted in a circular plane to determine the radial deformation of the cross section in different regions. Table 2 shows the radial strain of soil samples when the axial displacement reaches 12 mm.

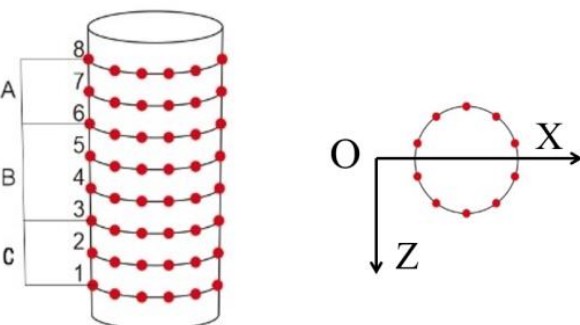

**Figure 5.** Schematic diagram of soil sample partition method.

**Table 2.** Radial strain of soil samples with an axial displacement of 12 mm.

| Dry Density | Chamber Pressure | Conventional Measurement | Photogrammetry | | |
| --- | --- | --- | --- | --- | --- |
| | | | Part A | Part B | Part C |
| g/cm$^3$ | kPa | % | % | % | % |
| 1.4 | 100 | 9.216 | 7.581 | 12.041 | 9.015 |
| 1.4 | 150 | 9.269 | 7.223 | 12.353 | 8.951 |
| 1.4 | 200 | 9.271 | 7.711 | 12.248 | 8.845 |
| 1.5 | 100 | 9.331 | 9.625 | 14.721 | 11.132 |
| 1.5 | 150 | 9.274 | 10.323 | 14.561 | 10.851 |
| 1.5 | 200 | 9.273 | 10.421 | 14.075 | 10.366 |

The comparison of the measured results of the two methods shows that the calculated results based on the area correction formula cannot fully reflect the actual radial deformation of soil samples. As the end restraint restricts the deformation of the soil sample, the radial deformation of area B in the middle of the soil sample is significantly greater than that of area A and area C, which proves the inconsistency of the deformation of the soil sample, indicating that the photogrammetric method has a good measurement effect on the uneven deformation of the soil sample. Hence, the results obtained from our method can more accurately reflect the shape of the soil sample.

### 4.2. Analysis of Stress–Strain Curve

The stress–strain curve is an important reference index for establishing soil sample constitutive model and studying soil shear band evolution. Since the deviatoric stress of the soil sample is related to the axial stress area, the obtained radial deformation data of the soil sample can be used to calculate the actual action area of the axial force at each loading stage, so as to obtain the axial deviatoric stress in different regions of the soil sample. Figure 6 demonstrates the stress–strain curves of the upper, middle and lower

regions of soil samples under 100 kPa and 200 kPa confining pressures and the stress–strain curves obtained by conventional measurement methods.

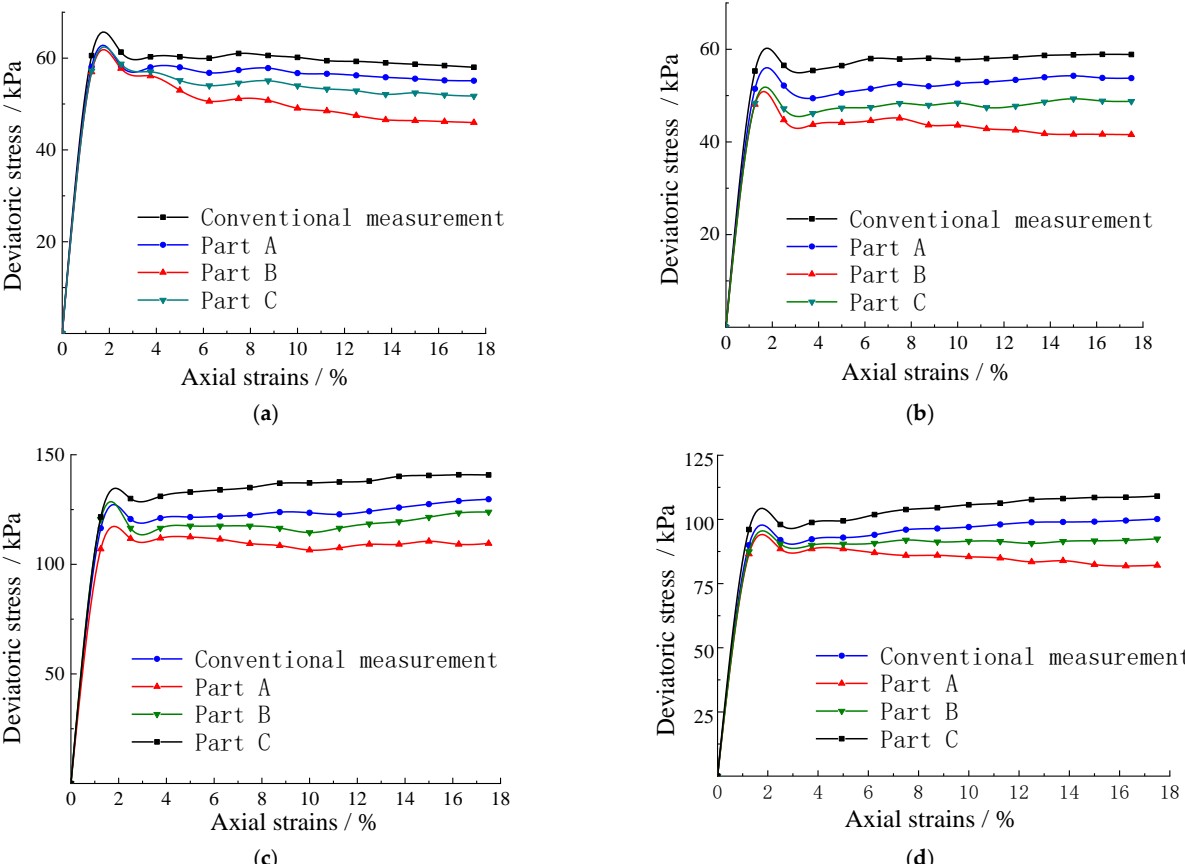

**Figure 6.** Comparison of stress–strain curves of soil samples under confining pressures of 100 kPa and 200 kPa. (**a**) 1.4 g/cm$^3$, 100 kPa; (**b**) 1.4 g/cm$^3$, 200 kPa; (**c**) 1.5 g/cm$^3$, 100 kPa; (**d**) 1.5 g/cm$^3$, 200 kPa.

At the initial stage of the test, the soil sample can be treated as a uniform unit, and the stress is relatively uniform. The action areas of soil axial force obtained by the conventional measurement method are roughly the same as that from the image measurement method, resulting in close stress–strain curves. With the increase of axial displacement, the stress on particles in different areas of the soil is no longer uniform, so the difference in the stress–strain curves from these two methods starts occurring, which is reflected in the fact that the local curves of soil samples measured by images are generally lower than those obtained by conventional measurements. The reason is that the conventional measurement method uses strain simplification to calculate the soil deformation, so the radial deformation obtained is the average radial deformation. As a result, the action area of the axial force is determined as smaller than the actual area, which overestimates the deviatoric stress of the soil sample. At the same time, the end restraint has a restrictive effect on the deformation of the upper (A) and lower (C) regions of the soil sample. As such, the axial stress area at both ends of the soil sample is smaller than the theoretical stress area. In contrast, region B of the soil sample is not affected by the end restraint, and its stress–strain curve is lower than that of other regions.

The shear band evolution process of soil samples can be studied from stress and strain aspects. It is generally believed that the shear band starts to occur when the stress reaches the peak value in the stress–strain curve, and the shear band achieves complete development when the last axial strain inflection point occurs [40]. Compared to the stress–strain curve measured by the conventional method, the peak stress in area B of the soil samples measured by the method of the present work is always lower, as shown in Figure 6,

indicating that the deviatoric stress in the middle of the soil sample measured by the new method is smaller when the shear band begins to form. When conventional measurement methods are used for measurement, shear bands appear before the soil sample reaches the peak stress and become more and more obvious with the increase of internal stress of the soil sample. Meanwhile, it is found that the occurrence of the last strain inflection in the B area measured by the new method appears later than that measured by the conventional method, which indicates that region B is not affected by the end restraint, and the large slip between particles in the soil sample results in the earlier start of the shear band evolution. Therefore, compared with the traditional measurement method, the method adopted in this work can more accurately measure the peak stress point and inflection point in the middle region of the soil sample, so it is easier to evaluate the formation and development of the shear band.

### 4.3. Dynamic Evolution Analysis of Shear Band Using Displacement Field

Although the stress–strain curve can be used to estimate the process of shear band formation, this method is not intuitive and accurate enough, nor can it help to understand the evolution information in the process of shear band formation. The local deformation of the soil sample in the deformation process can be determined by comparing the three-dimensional space coordinates before and after the deformation of the coding point. Many studies show that the local axial deformation of a soil sample can effectively reflect the formation and evolution of the shear band in the process of shear failure of the soil sample [41,42]. Figure 7 demonstrates the three-dimensional scattered points of three-dimensional coordinates ($X$, $Y$, $Z$) extracted from a soil sample with a dry density of 1.5 g/cm$^3$ and confining pressure of 200 kPa.

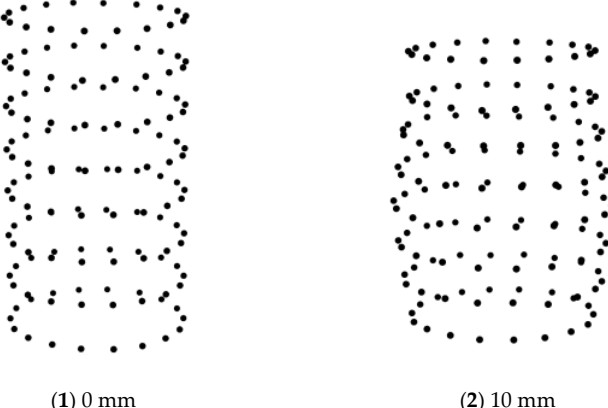

(**1**) 0 mm　　　　　　　　　　　　　　　(**2**) 10 mm

**Figure 7.** Three-dimensional scatter plots of the soil sample with different axial displacements.

It can be seen that the soil sample is a standard cylinder at the initial state and then becomes an elliptical cylinder protruding in the middle when the axial displacement reaches 10 mm. Although such a comparison can intuitively show the position change of each point in the soil sample, the dynamic change rule of the shear band in the soil sample is yet to be known. Therefore, we used Matlab to conduct a polynomial interpolation on the four-dimensional coordinates ($X$, $Y$, $Z$, $dz$) of the discrete data points and obtained the three-dimensional displacement field diagram of continuous soil samples as shown in Figure 8, where $dz$ represents the axial displacement increment of each point of the soil sample, and the colors depict the deformation degrees of local regions of the soil sample.

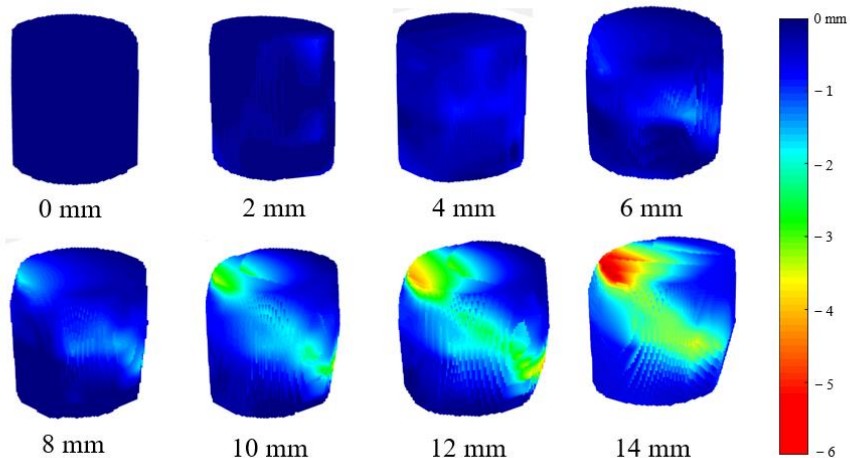

**Figure 8.** Three-dimensional displacement field of the soil sample (1.5 g/cm$^3$, 200 kPa).

In Figure 8, the three-dimensional spatial displacement field of the initial state of the soil sample has a uniform color distribution, which indicates that the confining pressure applied during the test does not form a shear effect on the soil sample. When the axial displacement reaches 2 mm, a compression effect starts to occur in the soil sample, but the axial and radial deformations are still relatively consistent, causing the soil shape to be nearly unchanged. At this time, the color distribution of the field map changes slightly, so it can be estimated as the initial stage of the soil sample shearing. When the axial displacement reaches 4 mm, the uneven particle distribution and density in the soil cause a large stress concentration effect and disordered deformation in the soil. From the initial stage of shearing, the color change of the displacement field is mainly concentrated in the middle area of the soil sample because the deformation of the upper and lower ends of the soil sample is limited by the end restraint and end friction, while the middle region of the soil sample is free from these. When the axial displacement reaches 6 mm, the local deformation area develops diagonally to form a connected area, and some local areas have undergone plastic deformation, which can be used to predict the development trend of the soil sample shear band. When the axial displacement reaches 12 mm, the deformation area in the middle region of the soil mass increases obviously, and the stress concentration effect in the plastic deformation area increases gradually. The uneven change areas in the soil sample gradually connect to form a significant band area, namely, the shear band of the soil sample. At this time, the shear band has completely penetrated the soil sample, and the soil has been sheared, resulting in the shear failure stage of the soil sample. After that, the soil sample is divided into upper and lower regions that are roughly symmetrical by the shear band, and the whole soil consists of independent upper and lower parts separated by the shear band, between which there is friction. With the continuous increase of the axial pressure, the axial displacement reaches 14 mm, and the upper and lower soil masses produce relative slip, leading to the post-failure stage. The deformation area of the soil mass has obvious signs of expansion centered on the shear band. Figure 9 compares the shear deformation of the soil sample with axial displacements of 10 mm and 14 mm.

The formation process of the shear band given by the displacement field method is consistent with the actual shear development process of the triaxial soil sample, proving that image measurement can clearly record the change process of the soil sample in a triaxial test and is capable of studying the non-uniformity of soil sample deformation while avoiding the defect of traditional triaxial test in measuring local changes of a soil sample. Moreover, the formation of the shear band can be visualized in the form of a displacement field graph with the new method, enabling the dynamic evolution law of the shear band of soil samples to be revealed in the triaxial test.

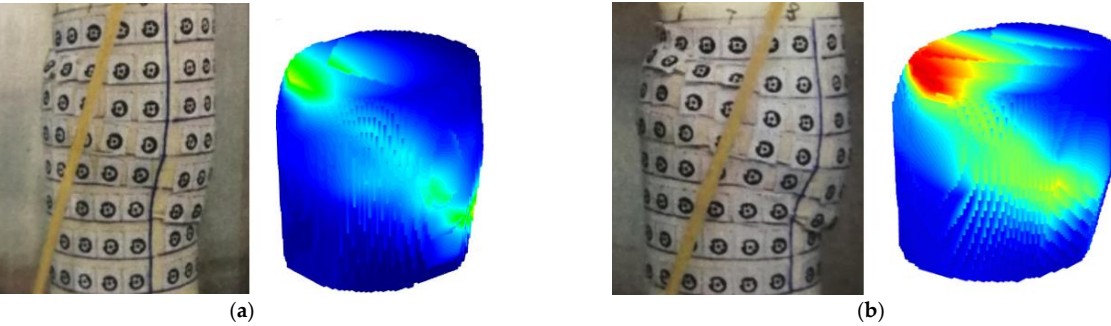

(**a**)  (**b**)

**Figure 9.** Comparison of the shear deformations under different axial displacements. (**a**) 10 mm axial displacement; (**b**) 14 mm axial displacement.

## 5. Conclusions

In this paper, digital image technology is applied to the undrained and unconsolidated triaxial test of soil samples. The stress–strain curve obtained from image measurement is compared with that obtained from the traditional triaxial test, and the dynamic evolution of the shear band of triaxial soil samples is studied based on the three-dimensional information of soil samples obtained from image measurement. The conclusions are as follows:

1.  The image measurement method is applied to the unconsolidated undrained test of Guilin red clay. The whole measurement process does not need to contact the soil sample and affect the normal data collection of the conventional triaxial test. This method mainly collects the deformation information of the pre-marked coding points on the soil sample surface to obtain the local deformation information, which is a challenge for the conventional triaxial test method.

2.  Digital image technology avoids the influence of end constraints in the triaxial test. The stress–strain curves of different regions of soil obtained by this method can be used to determine the formation process of shear bands in soil samples. By comparing the stress–strain curves measured by this method with those measured by conventional methods, it is known that the actual triaxial shear band occurs before the axial stress reaches the peak value. The stress–strain curve measured by the new method can more accurately reflect the time and process of shear failure of soil samples.

3.  Polynomial interpolation of the three-dimensional spatial coordinates of the code points on the surface of the soil sample can be transformed into three-dimensional spatial displacement fields, which enables observations of the formation, development and penetration of the soil sample shear band and therefore provides a new measurement method for the study of the formation and evolution of the soil shear band.

**Author Contributions:** Conceptualization, C.M. and H.W.; methodology, Y.X.; software, C.M. and Y.X.; validation, Y.X., W.L. and K.Y.; formal analysis, Y.X.; investigation, Y.X.; resources, C.M.; data curation, Y.X., W.L. and K.Y.; writing—original draft preparation, Y.X.; writing—review and editing, Y.X.; visualization, Y.X.; supervision, C.M.; project administration, H.W.; funding acquisition, C.M. All authors have read and agreed to the published version of the manuscript.

**Funding:** National Natural Science Foundation of China (41867039), National Natural Science Foundation of Guangxi (2020GXNSFAA297247), Project funded by the Southern Shishan Region Mine Geological Environment Restoration Engineering Technology Innovation Centre (cxzx2020-002).

**Institutional Review Board Statement:** Not applicable.

**Acknowledgments:** The authors gratefully acknowledge the supports from National Natural Science Foundation of China (41867039), National Natural Science Foundation of Guangxi (2020GXNS-FAA297247), Project funded by the Southern Shishan Region Mine Geological Environment Restoration Engineering Technology Innovation Centre (cxzx2020-002).

**Conflicts of Interest:** The authors declare no conflict of interest.

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
