# Peer review of "Study of Dynamic Evolution of the Shear Band in Triaxial Soil Samples Using Photogrammetry Technology"

_sustainability, doi:10.3390/su142114660_

Round 1

Reviewer 1 Report

The manuscript is very well written, it is acceptable for publications

Author Response

Thank you very much for your help in processing the review of our manuscript. We have now revised the manuscript to improve some of the manuscript's deficiencies. The manuscript has now been re-uploaded to the submission system.

Thank you again sincerely for your positive recognition and support of our manuscripts.

Reviewer 2 Report

In this manuscript, a measurement method that combines traditional photogrammetry with computer image processing technology was proposed. This research reflects the application of optical measurement in geotechnical engineering and has strong innovation. This manuscript can be accepted after minor revision:

1. Introduction: the author lists the previous research achievements in digital image technology. However, these research results are too old. It is suggested that the author refer to the latest research achievements in digital image technology:

Shape characteristics of coral sand from South China Sea. Journal of Marine Science and Engineering. 2020, 8(10): 803. DOI:10.3390/jmse8100803.

Experimental study on mesoscopic shear behavior of calcareous sand material with digital imaging approach. Advance in Civil Engineering. 2020, 8881264. DOI:10.1155/2020/8881264.

2. Section 2: This section describes the principle of digital image technology. Can the author briefly answer what are the key factors affecting the measurement accuracy of this test method? Although digital image technology is a relatively innovative test method, the accuracy of the test results using this method is the premise to ensure the accuracy of the data.

3. Line 141-143:With reference to Standard for Soil Test Methods (GB/T50123-2019), the static pressure sampling method was used to prepare soil samples with a dry density of 1.4 g/cm3, 1.5 g/cm3 and water content of 23% for standard triaxial tests. Why are the dry density and water content set to these values?

4. Line 147-148: Why does the author use unconsolidated untrained (UU) test to carry out this research? Why not CD test?

5.Line 170-172: ‘The deformation characteristics of the soil sample are featured with being big in the middle and small at both ends, based on which the test soil is divided into three regions (A, B and C) from top to bottom, as shown in Figure 5.’ The deformation characteristics of the specimen are not necessarily as you stated. It is suggested to change it to the following sentence:

Considering the influence of end restraint on specimen deformation, the test soil is divided into three regions (A, B and C) from top to bottom, as shown in Figure 5.

6. Line 202-204:‘the stress-strain curves from these two methods starts occurring, which is reflected in the fact that the local curves of soil samples measured by images are generally lower than those obtained by conventional measurements.’ What causes this result?

7. Section 4.3: Although the evolution process of the shear zone is a concern, the thickness of the shear zone cannot be ignored. Whether the thickness of shear band can be obtained by using this method? It is suggested that the author may try to explore the thickness of shear band and its influencing factors in future work.

Author Response

Thank you very much for your help in processing the review of our manuscript. The reviewers' comments have contributed greatly to improve the manuscript. We have carefully read the thoughtful comments from the reviewers and provided our point-by-point responses. The manuscript has now been revised through the 'track changes' function. Also, we try to address each point in detail below

Point 1: Introduction: the author lists the previous research achievements in digital image technology. However, these research results are too old. It is suggested that the author refer to the latest research achievements in digital image technology:

Shape characteristics of coral sand from South China Sea. Journal of Marine Science and Engineering. 2020, 8(10): 803. DOI:10.3390/jmse8100803.

Experimental study on mesoscopic shear behavior of calcareous sand material with digital imaging approach. Advance in Civil Engineering. 2020, 8881264. DOI:10.1155/2020/8881264.

Response 1: Thanks for the reviewer’s valuable comments. The article has been revised and the results of the work of Shen and Wang et al. are mentioned in the introduction section of the text.

Point 2: Section 2: This section describes the principle of digital image technology. Can the author briefly answer what are the key factors affecting the measurement accuracy of this test method? Although digital image technology is a relatively innovative test method, the accuracy of the test results using this method is the premise to ensure the accuracy of the data.

Response 2:  The test method used in this paper is influenced by the following factors:(1) light refractionï¼›(2) deformation of the pressure chamber at different envelope pressuresï¼›(3) the sharpness of the shots taken of the RAD code points, andï¼›(4) the number of images taken. Of these, the light refraction has the most significant effect on the test results.

Point 3: Line 141-143:‘With reference to Standard for Soil Test Methods (GB/T50123-2019), the static pressure sampling method was used to prepare soil samples with a dry density of 1.4 g/cm3, 1.5 g/cm3 and water content of 23% for standard triaxial tests.’ Why are the dry density and water content set to these values?

Response 3: From previous studies on Guilin red clay, it is clear that shear zones will appear in the soil samples when the "dry density is slightly below the maximum dry density and the moisture content is optimal", which facilitates the comparison between the shear zone model and the soil samples themselves in the later section.

Point 4: Line 147-148: Why does the author use unconsolidated untrained (UU) test to carry out this research? Why not CD test?

Response 4: This is because the main objective of this study was to try to complete the reduction of the shear zone with the aid of photogrammetry. The short time taken for the red clay UU test fits the needs of the experiment. In subsequent studies this will be attempted in the CD test.

Point 5: Line 170-172: ‘The deformation characteristics of the soil sample are featured with being big in the middle and small at both ends, based on which the test soil is divided into three regions (A, B and C) from top to bottom, as shown in Figure 5.’ The deformation characteristics of the specimen are not necessarily as you stated. It is suggested to change it to the following sentence:

Considering the influence of end restraint on specimen deformation, the test soil is divided into three regions (A, B and C) from top to bottom, as shown in Figure 5.

Response 5: Thank you for your suggestion, the article has been corrected in the corresponding place.

Point 6: Line 202-204:‘the stress-strain curves from these two methods starts occurring, which is reflected in the fact that the local curves of soil samples measured by images are generally lower than those obtained by conventional measurements.’ What causes this result?

Response 6: This is due to the fact that the radial deformation obtained in the conventional test is the average radial deformation after combining the uneven deformation of the end and middle of the soil sample, but the area of the axial force is smaller than the actual area of the soil sample, resulting in "the local curve of the soil sample from the image measurement is generally lower than the curve obtained from the conventional measurement".

Point 7: Section 4.3: Although the evolution process of the shear zone is a concern, the thickness of the shear zone cannot be ignored. Whether the thickness of shear band can be obtained by using this method? It is suggested that the author may try to explore the thickness of shear band and its influencing factors in future work.

Response 7: Thank you for your positive affirmation and support for our manuscripts. We also agree that the study of shear zone thickness is of great relevance. We will further try to extend the reduction and study of shear zone thicknesses.

Many thanks again to the Reviewers, their inputs are very helpful for improving the manuscript.

Reviewer 3 Report

This study combines photogrammetry and computer technology to apply to the unconsolidated undrained test of triaxial soil samples. The novel method can establish an intuitive shear band evolution model of soil samples. The authors should consider the following questions:

1.  Does the modified instrument will influence the test results when the coding points were applied in the membrane. You should add the comparison between the results, and point out the difference.

2. Why the unconsolidated undrained test was adopted with different confining pressures? The author should give the reason.

3.    In Fig. 6, the deviatoric stress should be the same for each tests, and why the values in each figure are not the same?

4.    In Fig. 6, the deviatoric stress shows reduction trend after the peak. But the some curves will increases with the displacement. If the shear band is formed, can the deviatoric stress increases again? Please check the values.

5.   Rui et al. (2021a,b, Acta Geotechnica) also recorded shear band formation during the (interface) shear test. The authors perhaps can consider to mention his work in the first 2 paragraph.

   If the above questions were well addressed, the paper can be considered to be accepted in the journal.

Author Response

Thank you very much for your help in processing the review of our manuscript. The reviewers' comments have contributed greatly to improve the manuscript. We have carefully read the thoughtful comments from the reviewers and provided our point-by-point responses. The manuscript has now been revised through the 'track changes' function. Also, we try to address each point in detail below

Point 1: Does the modified instrument will influence the test results when the coding points were applied in the membrane. You should add the comparison between the results, and point out the difference.

Response 1: The modified instrument does not affect the results of the code points on the membrane. This is because the effect of the modified instrument on the code points on the membrane is in the form of occlusion. In this paper, the occlusion problem is solved by photographing around the instrument, so that any occluded code points on the membrane in one direction will be filled in at other angles.

Point 2: Why the unconsolidated undrained test was adopted with different confining pressures? The author should give the reason.

Response 2: This is because the main objective of this study was to try to complete the reduction of the shear zone with the aid of photogrammetry. The short time consuming UU tests on red clay fit the needs of the test, so the UU tests were chosen to be carried out at different confining pressures.

Point 3: In Fig. 6, the deviatoric stress should be the same for each tests, and why the values in each figure are not the same?

Response 3: This is due to the fact that in Figure 6, (a)(b) compares (c)(d) using soil samples made at different dry densities, (a)(b) and (c)(d) were triaxially tested at different confining pressures and the different test conditions will result in different deviatoric stress results.

Point 4: In Fig. 6, the deviatoric stress shows reduction trend after the peak. But the some curves will increases with the displacement. If the shear band is formed, can the deviatoric stress increases again? Please check the values.

Response 4: This is due to the fact that under different test conditions, some specimens show strain hardening and the deviatoric stress increases continuously with increasing strain, with no peak point of deviatoric stress. Other curves show a decreasing trend, showing a strain softening and no further increase in the partial stress after the formation of the shear zone.

Point 5: Rui et al. (2021a,b, Acta Geotechnica) also recorded shear band formation during the (interface) shear test. The authors perhaps can consider to mention his work in the first 2 paragraph.

Response 5: Thank you for your suggestion, the article has been revised and the results of the work of Rui et al. are mentioned in the introduction section of the text.

Many thanks again to the Reviewers, their inputs are very helpful for improving the manuscript.

Round 2

Reviewer 3 Report

The paper can be accepted after addressing all comments. And now, point 1 and 5 of the comments is unsatisfied. In point 1, the author should add some solid comparison between the test results. 

Author Response

We are very sorry that our last response did not satisfy you. We have taken points 1 and 5 very seriously and have discussed them in depth, reworking our response after each comment.

Point 1: Does the modified instrument will influence the test results when the coding points were applied in the membrane. You should add the comparison between the results, and point out the difference.

Response 1: Thank you very much for your suggestion for our paper, but after careful consideration we have decided not to add the comparison between the results in this paper for the following reasons: The photogrammetry method used in this paper is based on the method proposed by Zhang in 2015(Zhang, X.; Li, L.; Chen, G.; Lytton, R. A Photogrammetry-Based Method to Measure Total and Local Volume Changes of Unsaturated Soils during Triaxial Testing. Acta Geotech. 2015, 10 (1), 55–82. https://doi.org/10.1007/s11440-014-0346-8.). Studies of the influence of this method on the measurements of triaxial tests are detailed in (Li, L.; Zhang, X. A New Triaxial Testing System for Unsaturated Soil Characterization. Geotech. Test. J. 2015, 38 (6), 20140201. https://doi.org/10.1520/GTJ20140201. Li, L.; Zhang, X.; Chen, G.; Lytton, R. Measuring Unsaturated Soil Deformations during Triaxial Testing Using a Photogrammetry-Based Method. Can. Geotech. J. 2016, 53 (3), 472–489. https://doi.org/10.1139/cgj-2015-0038.), photogrammetry can be used well in unsaturated soil tests and will not influence the test results. In the end, we have not added arguments to the text, considering that the main purpose of this paper is to extend the method and to complete the reduction of the soil shear zone.

Point 2: “Point 5” of the comments is unsatisfied.

Response 2: As a result of our checks, we have identified gaps in the references and have now added literature to the text [23][24]:

[23]Rui, S.; Wang, L.; Guo, Z.; Cheng, X.; Wu, B. Monotonic Behavior of Interface Shear between Carbonate Sands and Steel. Acta Geotech. 2021, 16 (1), 167–187. https://doi.org/10.1007/s11440-020-00987-9.

[24]Rui, S.; Wang, L.; Guo, Z.; Zhou, W.; Li, Y. Cyclic Behavior of Interface Shear between Carbonate Sand and Steel. Acta Geotech. 2021, 16 (1), 189–209. https://doi.org/10.1007/s11440-020-01002-x.

Thank you again sincerely for your positive recognition and support of our manuscripts.

Round 3

Reviewer 3 Report

Now, it can be accepted.